# Synthesis, Biological Evaluation and Docking Studies of Chalcone and Flavone Analogs as Antioxidants and Acetylcholinesterase Inhibitors

**Laura Díaz-Rubio [1]**, **Rufina Hernández-Martínez [2]**, **Arturo Estolano-Cobián [1]**, **Daniel Chávez-Velasco [3]**, **Ricardo Salazar-Aranda [4]**, **Noemí Waksman de Torres [4]**, **Ignacio A. Rivero [3]**, **Víctor García-González [5]**, **Marco A. Ramos [1],\*** and **Iván Córdova-Guerrero [1],\***

1 Facultad de Ciencias Químicas e Ingeniería, Universidad Autónoma de Baja California, Tijuana, B.C. 22390, México; ldiaz26@uabc.edu.mx (L.D.-R.); arturo.estolano@uabc.edu.mx (A.E.-C.)
2 Departamento de Microbiología, Centro de Investigación Científica y de Educación Superior de Ensenada, Ensenada, B.C. 22860, México; ruhernan@cicese.mx
3 Centro de Graduados e Investigación en Química, Tecnológico Nacional de México/Instituto Tecnológico de Tijuana, Tijuana, B.C. 22510, México; dchavez@tectijuana.mx (D.C.-V.); irivero@tectijuana.mx (I.A.R.)
4 Departamento de Química Analítica, Facultad de Medicina, Universidad Autónoma de Nuevo León, San Nicolas de los Garza, Monterrey 64460, México; salazar121212@yahoo.com (R.S.-A.); nwaksman@gmail.com (N.W.d.T.)
5 Facultad de Medicina, Universidad Autónoma de Baja California, Mexicali, B.C. 21000, México; vgarcia62@uabc.edu.mx
\* Correspondence: mramos@uabc.edu.mx (M.A.R.); icordova@uabc.edu.mx (I.C.-G.); Tel.: +52-664-120-7741 (I.C.-G.)

**Abstract:** Several oxidative processes are related to a wide range of human chronic and degenerative diseases, like Alzheimer's disease, which also has been related to cholinergic processes. Therefore, search for new or improved antioxidant molecules with acetylcholinesterase activity is essential to offer alternative chemotherapeutic agents to support current drug therapies. A series of chalcone (**2a–2k**) and flavone (**3a–3k**) analogs were synthesized, characterized, and evaluated as acetylcholinesterase (AChE) inhibitors, and antioxidant agents using 1,1-diphenyl-2-picrylhydrazyl (DPPH•), 2-2′-azino-bis-(3-ethylbenzothiazoline-6-sulfonate) (ABTS•), and β-carotene/linoleic acid bleaching assay. Compounds more active were **3j** and **2k** in DPPH with $EC_{50}$ of $1 \times 10^{-8}$ and $5.4 \times 10^{-3}$ μg/mL, respectively; **2g** and **3i** in ABTS ($1.14 \times 10^{-2}$ and $1.9 \times 10^{-3}$ μg/mL); **2e**, **2f**, **3f**, **2j**, and **3j** exceeded the α-tocopherol control in the β-carotene assay (98–99% of antioxidant activity). At acetylcholinesterase inhibition assay, flavones were more active than chalcones; the best results were compounds **2d** and **3d** ($IC_{50}$ 21.5 and 26.8 μg/mL, respectively), suggesting that the presence of the nitro group enhances the inhibitory activity. The docking of these two structures were made to understand their interactions with the AChE receptor. Although further in vivo testing must be performed, our results represent an important step towards the identification of improved antioxidants and acetylcholinesterase inhibitors.

**Keywords:** chalcone; flavone; antioxidant activities; DPPH; acetylcholinesterase; docking

## 1. Introduction

Collectively, radical and non-radical species formed by the partial reduction of oxygen are known as reactive oxygen species (ROS) [1]. When ROS overcame the cellular antioxidant defense system,

whether through an increase in ROS levels or a decrease in the cellular antioxidant capacity, oxidative stress occurs [2], which disturbs cellular metabolism, damages cellular constituents, and triggers the activation of specific signaling pathways [3]. The pathogenesis of several chronic and degenerative human diseases, including cardiovascular, neurodegenerative, and cancerous ones, has been linked to a cellular condition exhibiting oxidative stress [4,5]. To counteract this stressful condition, the human body has several mechanisms and produces a broad variety of antioxidants [6], which together quench ROS activities and reduce cellular damage [7].

Oxidative stress plays a crucial role in the etiology and pathogenesis of neurodegenerative diseases, like Alzheimer [8]. Alzheimer's disease (AD) is globally the most common cause of dementia in adults above 60 years [9]; ageing lowers the antioxidant systems activity, leading to free radical progressive accumulation, triggering lipid peroxidation mechanisms and structural damage to proteins and DNA [10], which leads to brain tissue damage. Its progression is associated to biochemical changes, like cholinergic deficit, neuronal metabolic damage derived from the glutamate excitotoxicity, and oxidative stress [11]. Acetylcholinesterase inhibitors (AChEI) are actually the best available pharmacotherapy for Alzheimer's disease symptoms treatment, increasing the levels of the acetylcholine neurotransmitter during the cerebral cortex synapses [12].

Flavonoids comprise a large family of plant-derived structurally-related polyphenolic compounds that are classified as anthocyanidins, flavonols, chalcones, aurones, flavanones, isoflavones, flavans, flavanonols, flavanols, and flavones [13]. Flavonoids have been found to display biological activities, such as anti-inflammatory [14], hepatoprotective [15], antiulcer [16], enzyme inhibitors, such as cyclooxygenase [17], xanthine oxidase [18], lipoxygenase [19], phosphodiesterase [20], as well as antiviral [21], cardioprotective [22], anticancer [23], and antioxidant [24] functions.

Pharmacological effects of flavonoids are related to their antioxidant activity, as they are powerful antioxidants against free radicals and ROS [12]. Structurally, the free radical scavenging capacity is primarily attributed to high reactivity of their hydroxyl substituents that participate in the reaction [25]. In this context, we report the chemical synthesis of chalcone and flavone analogs with different substituents on the respective B ring (Figure 1). In addition to the mandatory molecule characterization, their antioxidant and AChE inhibition activity was evaluated by in vitro testing, to get preliminary insights regarding the structure-activity relationship, allowing for us to further develop the continuous search for new therapeutic agents.

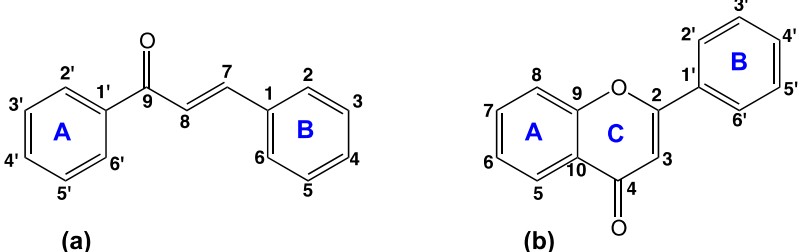

**Figure 1.** Basic structure and B ring position of chalcone (**a**) and flavone (**b**) molecules.

## 2. Materials and Methods

### 2.1. General Information

All commercial reagents and solvents were used as received and did not require any purification before use. Melting points were taken on a Mel-Temp melting point apparatus (Thermo Scientific). UV spectra were recorded on UV-VIS spectrophotometer model type Genesys 20 and expressed in nm. FT-IR spectroscopic studies were carried out on a FT-IR spectrophotometer Nicolet is5 (Thermo Scientific). NMR spectra were recorded on a Bruker Avance III spectrometer 400 MHz. The chemical shifts ($\delta$) are presented with tetramethylsilane (TMS) ($\delta$: 0.00) as the internal standard. Gas chromatography–Mass spectrometry data were recorded on a Thermo Scientific TRACE 1310 (GC)

and Thermo Scientific single quadrupole ISQ LT (MS), with a column model TG-SQC (30 m × 0.25 mm inner diameter, 0.25 μm film thickness). The detector temperature was 240 °C, the injector temperature was 250 °C, and transfer-line temperature was 250 °C; oven temperature started at 120 °C for 1 min, increased at a 40 °C/min rate until 280 °C, with a hold time of 10 min. Helium was employed as carrier gas, at 1 mL/min flow with split ratio 1:20. Column chromatography purifications were carried out on Silica Gel 60Å (Sigma-Aldrich, 230–400 mesh, Sigma-Aldrich, Milwaukee, WI, USA). The purity of compounds was checked by thin-layer chromatography (TLC) carried out on aluminum backed silica plates by Merck (Kenilworth, NJ, USA) and plates were revealed using a UV 254 nm light.

*2.2. Synthesis of Chalcone Derivatives*

To a mixture of 2-hydroxyacetophenone (1 Eq) and the appropriate benzaldehyde (**1a**–**1k**) (1 Eq) in ethanol/water (8:2) (10 mL), was added dropwise aqueous NaOH 50% (4 mL) at 0 °C. The reaction was stirred at room temperature till completion of reaction (monitored by TLC). The reaction solvent was completely evaporated to redissolve the crude in water and neutralize with a 10% HCl solution. The precipitate was filtered, washed with cold water, dried, and purified by column chromatography, employing eluent mixtures of n-hexane/ethyl acetate and dichloromethane/ethyl acetate in different proportions; some products were purified by recrystallization with a methanol/$H_2O$ (1:2) mixture. All of the structures were confirmed by mass and NMR spectra as discussed below.

(*E*)-1-(2-hydroxyphenyl)-3-phenylprop-2-en-1-one (**2a**): White solid (yield 46%). $C_{15}H_{12}O_2$. Mp = 58–60 °C. IR (ATR diamond, cm$^{-1}$) = 3358 (O-H), 3031 (C-H), 1683 (C=O) cm$^{-1}$. $^1$H NMR (400 MHz, CDCl$_3$) δ: 12.79 (s, 1H, OH), 7.93 (d, *J* = 15.3 Hz, 1H, H7), 7.92 (d, *J* = 8.1 Hz, 1H, H2′), 7.67 (d, *J* = 15.2 Hz, 1H, H8), 7.50-7.37 (m, 6H, H2, H3, H4, H5, H6, H4′), 7.06 (d, *J* = 8.6 Hz, 1H, H5′), 6.95 (d, *J* = 7.3 Hz, 1H, H3′). $^{13}$C NMR (100 MHz, CDCl$_3$) δ: 193.8 (s, C9), 163.7 (s, C6′), 145.5 (d, C7), 136.5 (d, C4′), 134.7 (s, C1), 131.0 (d, C2′), 129.7 (d, C3, C5), 129.1 (d, C2, C6), 128.7 (d, C4), 126.2 (d, C3′), 120.3 (s, C1′), 118.9 (d, C8), 118.7 (d, C5′). GC-MS m/z (rel. int.) = 224.15 [M]$^+$ (11), 223.16 (12), 147.08 (17), 120.06 (49), 104.11 (54), 92.05 (100), 77.08 (54).

(*E*)-1,3-bis(2-hydroxyphenyl)prop-2-en-1-one (**2b**): Yellow crystals (yield 41%). $C_{15}H_{12}O_3$. Mp = 137 °C. IR (ATR diamond, cm$^{-1}$) = 3267 (O-H), 3033 (C-H), 1683 (C=O), 1150 (C-O) cm$^{-1}$. $^1$H NMR (400 MHz, CDCl$_3$) δ: 12.63 (s, 1H, OH), 10.36 (s, 1H, OH), 8.21 (d, *J* = 15.3 Hz, 1H, H7), 8.21 (dd, $J_1$ = 7.9, $J_2$ = 1.4 Hz, 1H, H2′), 8.00 (d, *J* = 15.6 Hz, 1H, H8), 7.92 (dd, $J_1$ = 7.8, $J_2$ = 1.0 Hz, 1H, H6), 7.58 (ddd, $J_1$ = 8.2, $J_2$ = 8.6, $J_3$ = 1.4 Hz, 1H, H4′), 7.33 (ddd, $J_1$ = 8.2, $J_2$ = 8.4, $J_3$ = 1.3 Hz, 1H, H4), 7.03 (ddd, $J_1$ = 7.1 Hz, 1H, H5), 6.99 (dd, $J_1$ = 8.3 Hz, 1H, H5′), 6.92 (dd, $J_1$ = 7.5 Hz, 1H, H3′), 6.02 (ddd, $J_1$ = 8.3 Hz, 1H, H3). $^{13}$C NMR (100 MHz, CDCl$_3$) δ: 193.8 (s, C9), 157.4 (s, C2), 140.3 (d, C7), 136.0 (d, C4′), 132.4 (d, C2′), 130.5 (d, C4), 128.9 (d, C6), 121.1 (s, C1), 120.7 (d, C3′), 120.2 (s, C1′), 119.4 (d, C8), 119.0 (d, C5), 117.7 (d, C5′), 116.2 (d, C3). GC-MS m/z (rel. int.) = 240.18 [M]$^+$ (1), 222.18 (8), 147.11 (2), 121.09 (28), 92.1 (100), 63.07 (73), 39.06 (56).

(*E*)-3-(4-hydroxy-3-methoxyphenyl)-1-(2-hydroxyphenyl)prop-2-en-1-one (**2c**): Orange crystals (yield 42%). $C_{16}H_{14}O_4$. Mp = 120 °C. IR (ATR diamond, cm$^{-1}$) = 3377 (O-H), 3009 (C-H), 2965 (C-H), 1682 (C=O) cm$^{-1}$. $^1$H NMR (400 MHz, CDCl$_3$) δ: 12.90 (s, 1H, OH), 7.94 (dd, $J_1$ = 8.1, $J_2$ = 1.6 Hz, 1H, H2′), 7.93 (d, *J* = 15.2 Hz, 1H, H7), 7.52 (d, *J* = 15.6 Hz, 1H, H8), 7.51 (ddd, $J_1$ = 8.4, $J_2$ = 8.4, $J_3$ = 1.6 Hz, 1H, H4′), 7.14-6.93 (m, 5H, H2, H3, H6, H3′, H5′), 3.97 (s, 3H, H10). $^{13}$C NMR (100 MHz, CDCl$_3$) δ: 193.6 (s, C9), 163.5 (s, C6′), 148.7 (s, C3), 146.9 (s, C4), 145.8 (d, C7), 136.1 (d, C4′), 129.5 (d, C2′), 127.4 (s, C1), 123.7 (d, C6), 121.6 (d, C3′), 120.9 (s, C1′), 118.7 (d, C8), 118.1 (d, C5′), 117.6 (d, C5), 110.3 (d, C2), 56.1 (q, C10). GC-MS m/z (rel. int.) = 270.25 [M]$^+$ (4), 150.19 (28), 135.16 (29), 121.15 (16), 120.15 (26), 107.16 (33), 92.11 (100), 77.15 (68), 64.12 (68).

(*E*)-3-(2-hydroxy-5-nitrophenyl)-1-(2-hydroxyphenyl)prop-2-en-1-one (**2d**): Yellow powder (yield 46%). $C_{15}H_{11}NO_5$. Mp = 120 °C. IR (ATR diamond, cm$^{-1}$) = 3565 (O-H), 3068 (C-H), 1656 (C=O), 1578 (N-O),

1337 (N-O), cm$^{-1}$. $^1$H NMR (400 MHz, CDCl$_3$) δ: 12.44 (s, 1H, OH), 8.81 (d, *J* = 2.7 Hz, 1H, H6), 8.28 (dd, $J_1$ = 8.4, $J_2$ = 1.3 Hz, 1H, H4), 8.196 (d, *J* = 15.5 Hz, 1H, H7), 8.192 (dd, $J_1$ = 9.2, $J_2$ = 2.8 Hz, 1H, H2′), 8.10 (d, *J* = 15.7 Hz, 1H, H8), 7.60 (dd, $J_1$ = 7.4, $J_2$ = 1.4 Hz, 1H, H4′), 7.13 (d, *J* = 9.1 Hz, 1H, H3′), 7.04 (dd, $J_1$ = 7.8, $J_2$ = 1.0 Hz, 1H, H5′), 7.03 (d, *J* = 7.8 Hz, 1H, H3). $^{13}$C NMR (100 MHz, CDCl$_3$) δ: 193.5 (s, C9), 162.8 (s, C2), 161.8 (s, C6′), 140.0 (s, C5), 137.4 (d, C7), 136.2 (d, C4′), 130.9 (d, C2′), 127.3 (d, C4, C6), 124.5 (d, C3′), 123.2 (s, C1′), 121.7 (d, C8), 119.0 (d, C5′), 117.6 (s, C1), 116.6 (d, C3). GC-MS m/z (rel. int.) = 285.15 [M]$^+$ (1), 253.64 (1), 220.22 (4), 191.54 (3), 171.27 (6), 149.28 (6), 137.01 (16), 117.17 (18), 84.18 (51), 73.10 (32), 69.20 (41), 63.11 (49), 61.14 (41), 30.16 (100).

(*E*)-1-(2-hydroxyphenyl)-3-(4-methoxyphenyl)prop-2-en-1-one (**2e**): Yellow crystals (yield 40%). C$_{16}$H$_{14}$O$_3$. Mp = 76–78 °C. IR (ATR diamond, cm$^{-1}$) = 3082 (C-H), 2963 (C-H), 1685 (C=O), 1255 (C-O) cm$^{-1}$. $^1$H NMR (400 MHz, CDCl$_3$) δ: 12.91 (s, 1H, OH), 7.919 (dd, $J_1$ = 8.0, $J_2$ = 1.7 Hz, 1H, H2′), 7.911 (d, *J* = 15.0 Hz, 1H, H7), 7.62 (dd, $J_1$ = 8.8, $J_2$ = 1.7 Hz, 1H, H2), 7.60 (dd, $J_1$ = 8.5, $J_2$ = 1.7 Hz, 1H, H6), 7.54 (d, *J* = 15.0 Hz, 1H, H8), 7.49 (ddd, $J_1$ = 8.6, $J_2$ = 8.5, $J_3$ = 1.5 Hz, 1H, H4′), 7.02 (dd, $J_1$ = 8.4, $J_2$ = 1.0 Hz, 1H, H5′), 6.95 (dd, $J_1$ = 8.8, $J_2$ = 2.0 Hz, 2H, H3, H5), 6.94 (ddd, $J_1$ = 9.6, $J_2$ = 10.9, $J_3$ = 1.2 Hz, 1H, H3′), 3.85 (s, 3H, H10). $^{13}$C NMR (100 MHz, CDCl$_3$) δ: 193.8 (s, C9), 163.7 (s, C6′), 162.1 (s, C4), 145.4 (d, C7), 136. 2 (s, C4′), 130.6 (d, C2, C6), 129.6 (d, C2′), 127.5 (s, C1), 120.2 (s, C1′), 118.8 (d, C5′), 118.7 (d, C3′), 117.7 (d, C8), 114.6 (d, C3, C5), 55.5 (q, C10). GC-MS m/z (rel. int.) = 254.19 [M]$^+$ (17), 253.20 (12), 134.12 (100), 121.11 (34), 119.12 (40), 92.07 (79), 65.08 (84), 63.07 (90).

(*E*)-1-(2-hydroxyphenyl)-3-(3-methoxyphenyl)prop-2-en-1-one (**2f**): Orangish yellow crystals (yield 40%). C$_{16}$H$_{14}$O$_3$. Mp = 86–90 °C. IR (ATR diamond, cm$^{-1}$) = 3081 (C-H), 2918 (C-H), 1636 (C=O), 1253 (C-O) cm$^{-1}$. $^1$H NMR (400 MHz, CDCl$_3$) δ: 12.77 (s, 1H, OH), 7.91 (dd, $J_1$ = 8.0, $J_2$ = 1.6 Hz, 1H, H2′), 7.89 (d, *J* = 15.8 Hz, 1H, H7), 7.63 (d, *J* = 15.4 Hz, 1H, H8), 7.51 (ddd, $J_1$ = 8.5, $J_2$ = 8.6, $J_3$ = 1.6 Hz, 1H, H4′), 7.36 (dd, $J_1$ = 7.8 Hz, 1H, H6), 7.26 (dd, $J_1$ = 7.6 Hz, 1H, H4), 7.16 (dd, $J_1$ = 2.2, $J_2$ = 1.8 Hz, 1H, H2), 7.03 (dd, $J_1$ = 8.4, $J_2$ = 1.0 Hz, 1H, H5′), 6.99 (ddd, $J_1$ = 8.1, $J_2$ = 8.2, $J_3$ = 1.0 Hz, 1H, H5), 6.96 (ddd, $J_1$ = 8.1, $J_2$ = 8.2, $J_3$ = 1.1 Hz, 1H, H3′), 3.85 (s, 3H, H10). $^{13}$C NMR (100 MHz, CDCl$_3$) δ: 193.85 (s, C9), 163.7 (s, C6′), 160.1 (s, C3), 145.5 (d, C7), 136.5 (d, C4′), 136.13 (s, C1), 130.1 (d, C2′), 129.8 (d, C5), 121.4 (d, C3′), 120.6 (s, C1′), 120.1 (d, C6), 118.9 (d, C8), 118.7 (d, C5′), 116.7 (d, C4), 113.9 (d, C2), 55.5 (q, C10). GC-MS m/z (rel. int.) = 254.15 [M]$^+$ (11), 253.18 (8), 165.12 (3), 147.08 (27), 134.10 (17), 121.05 (52), 102.06 (18), 92.05 (36), 89.06 (41), 65.04 (100).

(*E*)-3-(3,4-dimethoxyphenyl)-1-(2-hydroxyphenyl)prop-2-en-1-one (**2g**): yellow crystals (yield 57%). C$_{17}$H$_{16}$O$_4$. Mp = 88–90 °C. IR (ATR diamond, cm$^{-1}$) = 3079 (C-H), 1694 (C=O), 1142 (C-O) cm$^{-1}$. $^1$H NMR (400 MHz, CDCl$_3$) δ: 12.90 (s, 1H, OH), 7.94 (dd, $J_1$ = 8.1, $J_2$ = 1.8 Hz, 1H, H2′), 7.89 (d, *J* = 15.4 Hz, 1H, H7), 7.53 (d, *J* = 15.3 Hz, 1H, H8), 7.50 (ddd, $J_1$ = 8.4, $J_2$ = 8.5, $J_3$ = 1.3, 1H, H4′), 7.27 (dd, $J_1$ = 8.3, $J_2$ = 2.0 Hz, 1H, H6), 7.17 (d, 1H, H2), 7.03 (ddd, $J_1$ = 8.3, $J_2$ = 8.3, $J_3$ = 1.0 Hz, 1H, H3′), 6.95 (dd, $J_1$ = 8.1, $J_2$ = 1.0 Hz, 1H, H5′), 6.91 (d, 1H, H5), 3.96 (s, 3H, H10), 3.93 (s, 3H, H11). $^{13}$C NMR (100 MHz, CDCl$_3$) δ: 193.7 (s, C9), 163.6 (s, C6′), 151.9 (s, C3), 149.4 (s, C4), 145.7 (d, C7), 136.2 (d, H4′), 129.6 (d, C2′), 127.7 (s, C1), 123.6 (d, C6), 121.6 (d, C3′), 120.2 (s, C1′), 118.8 (d, C8), 117.9 (d, C5′), 111.3 (d, C5), 110.5 (d, C2), 56.1 (q, C10, C11). GC-MS m/z (rel. int.) = 284.20 [M]$^+$ (8), 165.16 (4), 164.16 (33), 151.15 (25), 121.13 (24), 103.11 (27), 92.06 (69), 77.08 (100).

(*E*)-3-(4-chlorophenyl)-1-(2-hydroxyphenyl)prop-2-en-1-one (**2h**): yellow crystals (yield 97%). C$_{15}$H$_{11}$ClO$_2$. Mp = 70–73 °C. IR (ATR diamond, cm$^{-1}$) = 3065 (C-H), 1641 (C=O), 748 (C-Cl) cm$^{-1}$. $^1$H NMR (400 MHz, CDCl$_3$) δ: 12.72 (s, 1H, OH), 7.90 (dd, $J_1$ = 8.0, $J_2$ = 1.6 Hz, 1H, H2′), 7.86 (d, *J* = 15.4 Hz, 1H, H7), 7.62 (d, *J* = 15.4 Hz, 1H, H8), 7.59 (d, 2H, H2, H6), 7.51 (ddd, $J_1$ = 8.8, $J_2$ = 8.6, $J_3$ = 1.5 Hz, 1H, H4′), 7.41 (d, 2H, H3, H5), 7.03 (dd, $J_1$ = 8.4, $J_2$ = 1.0 Hz, 1H, H5′), 6.95 (ddd, $J_1$ = 8.1, $J_2$ = 8.0, $J_3$ = 1.1 Hz, 1H, H3′). $^{13}$C NMR (100 MHz, CDCl$_3$) δ: 193.5 (s, C9), 163.7 (s, C6′), 144.0 (d, C7), 137.0 (d, C4′), 136.6 (s, C1), 133.2 (s, C4), 129.9 (d, C2, C6), 129.7 (d, C2′), 129.4 (d, C3, C5), 120.7 (d,

C3′), 120, 0 (s, C1′), 119.0 (d, C8), 118.8 (d, C5′). GC-MS m/z (rel. int.) = 258.21 [M]$^+$ (4), 224.12 (5), 223.18 (26), 165.18 (7), 147.11 (20), 121.13 (32), 101.10 (39), 93.13 (26), 65.12 (100).

(*E*)-3-(4-(dimethylamino)phenyl)-1-(2-hydroxyphenyl)prop-2-en-1-one (**2i**): purple red crystals (yield 51%). C$_{17}$H$_{17}$NO$_2$. Mp = 163–165 °C. IR (ATR diamond, cm$^{-1}$) = 3064 (C-H), 1661 (C=O), 1276 (C-N) cm$^{-1}$. $^1$H NMR (400 MHz, CDCl$_3$) δ: 13.18 (s, 1H, OH), 7.919 (d, *J* = 14.9 Hz, 1H, H7), 7.915 (dd, *J*$_1$ = 8.3, *J*$_2$ = 1.8 Hz, 1H, H2′), 7.56 (dd, *J*$_1$ = 8.8, *J*$_2$ = 1.8 Hz, 2H, H2, H6), 7.46 (ddd, *J*$_1$ = 8.5, *J*$_2$ = 8.2, *J*$_3$ = 1.6 Hz, 1H, H4′), 7.45 (d, *J* = 15.1 Hz, 1H, H8), 7.00 (dd, *J*$_1$ = 8.3, *J*$_2$ = 1.0 Hz, 1H, H5′), 6.92 (ddd, *J*$_1$ = 8.2, *J*$_2$ = 8.0, *J*$_3$ = 1.2 Hz, 1H, H3′), 6.68 (dd, *J*$_1$ = 8.9, *J*$_2$ = 2.8 Hz, 2H, H3, H5), 3.03 (s, 6H, H10). $^{13}$C NMR (100 MHz, CDCl$_3$) δ: 193.6 (s, C9), 163.6 (s, C6′), 152.4 (s, C4), 146.6 (d, C7), 135.7 (d, C4′), 130.9 (d, C2, C6), 129.4 (d, C2′), 122.4 (s, C1), 120.5 (d, C3′), 118.6 (s, C1′), 118.5 (d, C8), 114.4 (d, C5′), 111.9 (d, C3, C5), 40.1 (q, C10). GC-MS m/z (rel. int.) = 267.11 [M]$^+$ (6), 266.11 (3), 220.13 (9), 165.16 (8), 152.16 (3), 104.05 (11), 92.09 (21), 89.06 (39), 76.07 (100).

(*E*)-3-(anthracen-9-yl)-1-(2-hydroxyphenyl)prop-2-en-1-one (**2j**): orange crystals (yield 45%). C$_{23}$H$_{16}$O$_2$. Mp = 148-150 °C. IR (ATR diamond, cm$^{-1}$) = 3046 (C-H), 1633 (C=O), 1157 (C-O) cm$^{-1}$. $^1$H NMR (400 MHz, CDCl$_3$) δ: 12.85 (s, 1H, OH), 8.89 (d, *J* = 15.6 Hz, 1H, H15), 8.45 (s, 1H, H8), 8.27 (dd, *J*$_1$ = 9.4, *J*$_2$ = 2.9 Hz, 2H, H3, H13), 8.02 (dd, *J*$_1$ = 9.6, *J*$_2$ = 2.9 Hz, 2H, H6, H10), 7.81 (dd, *J*$_1$ = 9.6, *J*$_2$ = 1.4 Hz, 1H, H2′), 7.65 (d, *J* = 15.6 Hz, 1H, H16), 7.53-7.46 (m, 4H, H4, H5, H11, H12), 7.53 (ddd, *J*$_1$ = 8.3, *J*$_2$ = 8.3, *J*$_3$ = 1.7 Hz, H4′) 7.08 (dd, *J*$_1$ = 8.4, *J*$_2$ = 1.0 Hz, 1H, H5′), 6.89 (ddd, *J*$_1$ = 8.1, *J*$_2$ = 8.1, *J*$_3$ = 1.1 Hz, 1H, H3′). $^{13}$C NMR (100 MHz, CDCl$_3$) δ: 193.3 (s, C17), 163.9 (s, C6′), 142.7 (d, C15), 136.7 (d, C4′), 131.4 (s, C1), 130.0 (s, C7, C9), 129.8 (d, C6, C10, C2′), 129.5 (d, C3, C13), 129.1 (s, C2, C14), 128.9 (d, C4, C12), 126.7 (d, C5, C11), 125.6 (s, C3′), 120.1 (d, C8, C4′), 119.1 (d, C16), 118.8 (d, C5′). GC-MS m/z (rel. int.) = 324.22 [M]$^+$(27), 281.96 (5), 268.48 (5), 203.17 (17), 202.14 (35), 200.18 (11), 178.27 (12), 124.79 (15), 121.01 (90), 93.12 (36), 65.05 (100).

(*E*)-3-(furan-2-yl)-1-(2-hydroxyphenyl)prop-2-en-1-one (**2k**): yellow crystals (yield 84%). C$_{13}$H$_{10}$O$_3$. Mp = 90–92 °C. IR (ATR diamond, cm$^{-1}$) = 3128-3150 (C-H), 1669 (C=C), 1636 (C=O), 1158 (C-O) cm$^{-1}$. $^1$H NMR (400 MHz, CDCl$_3$) δ: 12.86 (s, 1H, OH), 7.91 (dd, *J*$_1$ = 8.0, *J*$_2$ = 1.4 Hz, 1H, H2′), 7.69 (d, *J*= 15.1 Hz, 1H, H5), 7.56 (d, *J* = 15.1 Hz, 1H, H6), 7.55 (dd, *J*$_1$ = 3.4, *J*$_2$ = 1.3 Hz, 1H, H4), 7.49 (ddd, *J*$_1$ = 8.5, *J*$_2$ = 8.5, *J*$_3$ = 1.5 Hz, 1H, H4′), 6.94 (ddd, *J*$_1$ = 8.1, *J*$_2$ = 8.1, *J*$_3$ = 1.0 Hz, 1H, H3′), 6.76 (d, 1H, H2). $^{13}$C NMR (100 MHz, CDCl$_3$) δ: 193.4 (s, C7), 163.7 (s, C6′), 151.6 (s, C1), 145.5 (d, C4), 136.4 (d, C4′), 131.2 (d, C2′), 129.7 (d, C6), 120.2 (d, C3′), 118.9 (s, C1′), 118.6 (d, C5), 117.8 (d, C5′), 117.1 (d, C2), 113.0 (d, C3). GC-MS m/z (rel. int.) = 214.20 [M]$^+$ (6), 157.12 (3), 131.17 (3), 128.16 (3), 121.12 (23), 94.13 (18), 92.11 (13), 77.10 (9), 65.11 (74), 39.09 (100).

## 2.3. Synthesis of Flavone Derivatives

To a solution of the corresponding 2-hydroxychalcone (**2a–2k**) (1 Eq) and I$_2$ (1 Eq), was added DMSO (5 mL) and the reaction mixture was heated in an oil bath at 130 °C until the completion of reaction (monitored by TLC). After cooling, the reaction mixture was diluted with water and the iodine was removed by washing with a saturated solution of sodium thiosulfate. The products (**3a–3k**) were then extracted with ethyl acetate and purified by column chromatography, employing eluent mixtures of n-hexane/ethyl acetate, dichloromethane/ethyl acetate, and ethyl acetate/methanol in different proportions; some products were purified by recrystallization with a methanol/H$_2$O (1:2) mixture.

2-phenyl-4H-chromen-4-one (**3a**): white crystals (yield 61%). C$_{15}$H$_{10}$O$_2$. Mp = 85–87 °C. IR (ATR diamond, cm$^{-1}$) = 3072 (C-H), 1639 (C=O), 1030 (C-O) cm$^{-1}$. $^1$H NMR (400 MHz, CDCl$_3$) δ: 8.25 (dd, *J*$_1$ = 7.9, *J*$_2$ = 1.5 Hz, 1H, H5), 7.94 (dd, *J*$_1$ = 9.6, *J*$_2$ = 1.7 Hz, 2H, H2′, H6′), 7.72 (ddd, *J*$_1$ = 8.5, *J*$_2$ = 8.6, *J*$_3$ = 1.6 Hz, 1H, H7), 7.58-7.50 (m, 4H, H6, H8, H3′, H5′), 7.44 (ddd, *J*$_1$ = 8.0, *J*$_2$ = 8.0, *J*$_3$ = 1.0 Hz, 1H, H4′), 6.82 (s, 1H, H3). $^{13}$C NMR (100 MHz, CDCl$_3$) δ: 178.5 (s, C4), 163.5 (s, C2), 156.4 (s, C9), 133.8 (d, C7), 131.9 (s, C1′), 131.7 (d, C3′), 129.1 (d, C5′), 126.4 (d, C4′), 125.8 (d, C5, C2′, C6′), 125.3 (s, C10), 124.1 (d,

C6), 118.2 (d, C8), 107.7 (d, C3). GC-MS m/z (rel. int.) = 222.12 [M]$^+$ (19), 221.13 (5), 194.11 (14), 165.11 (7), 120.04 (48), 102.06 (28), 97.07 (15), 92.05 (100).

2-(2-hydroxyphenyl)-4H-chromen-4-one (**3b**): Pale yellow powder (yield 60%). $C_{15}H_{10}O_3$. Mp = 254–256 °C. IR (ATR diamond, cm$^{-1}$) = 3141 (O-H), 3098 (C-H), 1638 (C=O), 1051 (C-O) cm$^{-1}$. $^1$H NMR (400 MHz, CDCl$_3$) δ: 11.02 (s, 1H, OH), 8.17 (dd, 1H, H5), 8.05 (ddd, $J_1$ = 8.6, $J_2$ = 8.6, $J_3$ = 1.1 Hz, 1H, H7), 7.843 (ddd, $J_1$ = 7.8, $J_2$ = 8.0, $J_3$ = 1.0 Hz, 1H, H6), 7.845 (dd, $J_1$ = 8.4, $J_2$ = 1.4 Hz, 1H, H8), 7.80 (ddd, $J_1$ = 8.0, $J_2$ = 8.0, $J_3$ = 1.4 Hz, 1H, H4′), 7.70 (dd, $J_1$ = 8.6, $J_2$ = 2.3 Hz, 1H, H6′), 7.51 (ddd, $J_1$ = 7.9, $J_2$ = 8.0, $J_3$ = 1.9 Hz, 1H, H5′), 7.12 (s, 1H, H3), 6.92 (dd, 1H, H3′). $^{13}$C NMR (100 MHz, CDCl$_3$) δ: 177.0 (s, C4), 159.1 (s, C2), 156.7 (s, C9), 156.3 (s, C2′), 140.6 (d, C7), 136.2 (d, C4′), 134.1 (d, C6′), 125.2 (d, C5), 124.64 (s, C10), 124.61 (d, C6), 123.0 (d, C5′), 120.3 (s, C1′), 119.5 (d, C3′), 118.5 (d, C8), 111.4 (s, C3). GC-MS m/z (rel. int.) = 238.23 [M]$^+$ (10), 210.20 (2), 152.27 (3), 121.15 (17), 120.14 (10), 118.16 (12), 92.14 (61), 90.16 (30), 76.15 (21), 64.14 (59), 63.13 (100).

2-(4-hydroxy-3-methoxyphenyl)-4H-chromen-4-one (**3c**): Yellow powder (yield 40%). $C_{16}H_{12}O_4$. Mp = 198–200 °C. IR (ATR diamond, cm$^{-1}$) = 3295 (O-H), 3092 (C-H), 2935 (C-H), 1624 (C=O) cm$^{-1}$. $^1$H NMR (400 MHz, CDCl$_3$) δ: 8.12 (dd, $J_1$ = 8.0, $J_2$ = 1.6 Hz, 1H, H5), 8.03 (dd, $J_2$ = 2 Hz, 1H, H8), 7.82 (ddd, $J_1$ = 8.8, $J_2$ = 8.4, $J_3$ = 1.6 Hz, 1H, H7), 7.74 (ddd, $J_1$ = 8.4, $J_2$ = 8.4, $J_3$ = 1 Hz, 1H, H6), 7.59 (dd, $J_2$ = 2 Hz, 1H, H6′), 7.50 (dd, $J_1$ = 8.8, $J_2$ = 1.2 Hz, 1H, H2′), 6.98 (d, 1H, H3′), 6.84 (s, 1H, H3), 4.01 (s, 3H, H11). $^{13}$C NMR (100 MHz, CDCl$_3$) δ: 179.5 (s, C4), 163.9 (s, C2), 157.4 (s, C9), 151.5 (s, C4′), 148.3 (s, C5′), 135.3 (d, C7), 130.4 (d, C5), 126.5 (s, C10), 126.0 (d, C6), 125.4 (s, C1′), 124.6 (d, C2′), 119.4 (d, C3′), 110.4 (d, C8), 106.8 (d, C6′), 83.7 (d, C3), 57.0 (q, C11). GC-MS m/z (rel. int.) = 268.24 [M]$^+$ (20), 197.16 (8), 148.18 (16), 121.13 (27), 105.14 (47), 92.12 (95), 63.13 (100).

2-(2-hydroxy-5-nitrophenyl)-4H-chromen-4-one (**3d**): Pale yellow powder (yield 58%). $C_{15}H_9NO_5$. Mp = 268–270 °C. IR (ATR diamond, cm$^{-1}$) = 3298 (O-H), 3024 (C-H), 1652 (C=O), 1342 (N-O) cm$^{-1}$. GC-MS m/z (rel. int.) = 282.23 [M]$^+$ (3), 281.20 (10), 235.18 (3), 179.25 (7), 150.19 (10), 104.12 (9), 92.15 (8), 30.09 (100).

2-(4-methoxyphenyl)-4H-chromen-4-one (**3e**): white powder (yield 50%). $C_{16}H_{12}O_3$. Mp = 158–160 °C. IR (ATR diamond, cm$^{-1}$) = 3051 (C-H), 2920 (C-H), 1644 (C=O), 1024 (C-O) cm$^{-1}$. $^1$H NMR (400 MHz, CDCl$_3$) δ: 8.22 (dd, $J_1$ = 7.9, $J_2$ = 1.5 Hz, 1H, H5), 7.87 (dd, $J_1$ = 9 Hz, 2H, H2′, H6′), 7.69 (ddd, $J_1$ = 8.5, $J_2$ = 8.7, $J_3$ = 1.7 Hz, 1H, H7), 7.54 (dd, $J_1$ = 8.4 Hz, 1H, H8), 7.41 (ddd, $J_1$ = 8.0, $J_2$ = 8.0, $J_3$ = 1.0 Hz, 1H, H6), 7.02 (dd, $J_1$ = 9 Hz, 2H, H3′, H5′), 6.73 (s, 1H, H3), 3.87 (s, 3H, H11). $^{13}$C NMR (100 MHz, CDCl$_3$) δ: 178.4 (s, C4), 163.5 (s, C2), 162.5 (s, C4′), 156.3 (s, C9), 133.6 (d, C7), 128.1 (d, C2′, C6′), 125.7 (d, C5), 125.1 (d, C6), 124.1 (s, C10), 124.0 (s, C1′), 118.0 (d, C8), 114.5 (d, C3′, C5′), 106.3 (d, C3), 55.6 (q, C11). GC-MS m/z (rel. int.) = 252.18 [M]$^+$ (6), 209.14 (4), 181.09 (2), 132.11 (13), 117.09 (11), 92.05 (64), 89.07 (51), 76.05 (24), 63.07 (100).

2-(3-methoxyphenyl)-4H-chromen-4-one (**3f**): white powder (yield 49%). $C_{16}H_{12}O_3$. Mp = 118–120 °C. IR (ATR diamond, cm$^{-1}$) = 3036 (C-H), 2844 (C-H), 1639 (C=O), 1130 (C-O) cm$^{-1}$. $^1$H NMR (400 MHz, CDCl$_3$) δ: 8.24 (dd, $J_1$ = 8.0, $J_2$ = 1.2 Hz, 1H, H5), 7.71 (ddd, $J_1$ = 8.4, $J_2$ = 8.8, $J_3$ = 1.6 Hz, 1H, H5′), 7.57 (dd, $J_1$ = 8.8 Hz, 1H, H8), 7.52 (ddd, $J_1$ = 8.8, $J_2$ = 8.8, $J_3$ = 1.2 Hz, 1H, H7), 7.44 (ddd, $J_1$ = 8.0, $J_2$ = 2.8, $J_3$ = 2.0 Hz, 2H, H2′, H6′), 7.42 (dd, $J_1$ = 8.0, $J_2$ = 2.8 Hz, 1H, H4′), 7.08 (ddd, $J_1$ = 8.4, $J_2$ = 8.0, $J_3$ = 1.0 Hz, 1H, H6), 6.83 (s, 1H, H3), 3.89 (s, 3H, H11). $^{13}$C NMR (100 MHz, CDCl$_3$) δ: 178.6 (s, C4), 163.3 (s, C2), 160.1 (s, C3′), 156.3 (s, C9), 133.9 (d, C7), 133.2 (s, C1′), 130.2 (d, C5′), 125.8 (d, C5), 125.3 (s, C10), 124.1 (d, C6), 118.8 (d, C6′), 118.2 (d, C8), 117.3 (d, C4′), 111.9 (d, C2′), 107.9 (d, C3), 55.6 (q, C11). GC-MS m/z (rel. int.) = 252.25 [M]$^+$ (32), 251.23 (6), 224.24 (13), 152.19 (12), 132.18 (81), 120.14 (44), 102.17 (63), 92.13 (100).

2-(3,4-dimethoxyphenyl)-4H-chromen-4-one (**3g**): white powder (yield 85%). $C_{17}H_{14}O_4$. Mp = 128–130 °C. IR (ATR diamond, cm$^{-1}$) = 3062 (C-H), 2929 (C-H), 1651 (C=O), 1144 (C-O) cm$^{-1}$. $^1$H NMR (400 MHz, CDCl$_3$) δ: 8.23 (dd, $J_1$ = 7.9, $J_2$ = 1.5 Hz, 1H, H5), 7.70 (ddd, $J_1$ = 8.5, $J_2$ = 8.6, $J_3$ = 1.7 Hz, 1H, H7), 7.57 (dd, $J_1$ = 8.4, $J_2$ = 1.9 Hz, 2H, H8, H2′), 7.42 (ddd, $J_1$ = 8.0, $J_2$ = 7.6, $J_3$ = 1.0 Hz, 1H, H6), 7.39 (ddd, 1H, H6′), 6.99 (d, 1H, H3′), 6.77 (s, 1H, H3), 3.98 (s, 3H, H11), 3.96 (s, 3H, H12). $^{13}$C NMR (100 MHz, CDCl$_3$) δ: 193.7 (s, C4), 163.6 (s, C2), 151.9 (s, C9), 149.4 (s, C5′), 145.7 (s, C4′), 136.2 (d, C7), 129.6 (d, C5), 127.7 (s, C10), 123.6 (d, C6), 121.6 (s, C1′), 120.2 (d, C2′), 118.8 (d, C8), 117.9 (d, C3′), 111.3 (d, C6′), 110.5 (d, C3), 56.1 (q, C11, C12). GC-MS m/z (rel. int.) = 282.18 [M]$^+$ (35), 168.13 (12), 147.11 (16), 127.18 (25), 121.08 (100), 119.05 (32), 92.07 (86), 91.09 (99), 76.08 (91).

2-(4-chlorophenyl)-4H-chromen-4-one (**3h**): white powder (yield 82%). $C_{15}H_9ClO_2$. Mp = 188–190 °C. IR (ATR diamond, cm$^{-1}$) = 3071 (C-H), 1647 (C=O), 756 (C-Cl) cm$^{-1}$. $^1$H NMR (400 MHz, CDCl$_3$) δ: 8.23 (dd, $J_1$ = 7.9, $J_2$ = 1.4 Hz, 1H, H5), 7.86 (dd, $J_1$ = 8.8 Hz, 2H, H3′, H5′), 7.71 (ddd, $J_1$ = 8.5, $J_2$ = 8.7, $J_3$ = 1.7 Hz, 1H, H7), 7.55 (dd, $J_1$ = 8.4, $J_2$ = 1.0 Hz, 1H, H8), 7.50 (dd, $J_1$ = 8.8 Hz, 2H, H2′, H6′), 7.43 (ddd, $J_1$ = 8.0, $J_2$ = 7.8, $J_3$ = 1.0 Hz, 1H, H6), 6.77 (s, 1H, H3). $^{13}$C NMR (100 MHz, CDCl$_3$) δ: 178.2 (s, C4), 162.3 (s, C2), 156.2 (s, C9), 138.0 (d, C7), 134.0 (s, C4′), 130.3 (d, C2′, C6′), 129.4 (d, C3′, C5′), 127.6 (s, C1′), 125.8 (d, C5), 125.4 (s, C10), 124.0 (d, C6), 118.1 (d, C8), 107.8 (d, C3). GC-MS m/z (rel. int.) = 258.18 [M]$^+$ (3), 256.18 (10), 228.16 (6), 165.18 (3), 136.13 (13), 120.12 (29), 101.13 (28), 92.11 (100).

2-(4-(dimethylamino)phenyl)-4H-chromen-4-one (**3i**): white powder (yield 67%). $C_{17}H_{15}NO_2$. Mp = 193–195 °C. IR (ATR diamond, cm$^{-1}$) = 3057 (C-H), 1659 (C=O), 1277 (C-N) cm$^{-1}$. $^1$H NMR (400 MHz, CDCl$_3$) δ: 8.17 (dd, $J_1$ = 8.8, $J_2$ = 2 Hz, 1H, H5), 8.04-7.78 (m, 3H, H6, H7, H8), 7.77 (dd, 2H, H3′, H5′), 6.94 (dd, 2H, H2′, H6′), 6.84 (s, 1H, H3), 2.99 (s, 6H, H11). $^{13}$C NMR (100 MHz, CDCl$_3$) δ: 176.63 (s, C4), 162.4 (s, C2), 155.4 (s, C9), 152.5 (s, C4′), 135.7 (d, C7), 133.8 (d, C2′), 133.5 (d, C6′), 133.1 (d, C5), 125.2 (s, C10), 124.6 (d, C6), 118.2 (s, C1′), 117.9 (d, C8), 116.7 (d, C3′), 103.9 (d, C3). GC-MS m/z (rel. int.) = 265.26 [M]$^+$ (10), 261.68 (1), 195.10 (2), 104.19 (10), 90.71 (9), 74.87 (11), 73.05 (100).

2-(anthracen-9-yl)-4H-chromen-4-one (**3j**): yellow powder (yield 84%). $C_{23}H_{14}O_2$. Mp = 193–195 °C. IR (ATR diamond, cm$^{-1}$) = 3063 (C-H), 1643 (C=O), 1122 (C-O) cm$^{-1}$. $^1$H NMR (400 MHz, CDCl$_3$) δ: 8.62 (s, 1H, H8′), 8.40 (dd, $J_1$ = 7.96, $J_2$ = 1.6 Hz, 1H, H5), 8.08 (dd, $J_1$ = 6.7, $J_2$ = 1.5 Hz, 2H, H3′, H6′), 7.97 (dd, $J_1$ = 7.9, $J_2$ = 1.9 Hz, 2H, H10′, H13′), 7.73 (ddd, $J_1$ = 8.5, $J_2$ = 8.6, $J_3$ = 1.6 Hz, 1H, H7), 7.53-7.47 (m, 6H, H6, H8, H4′, H5′, H11′, H12′), 6.69 (s, 1H, H3). $^{13}$C NMR (100 MHz, CDCl$_3$) δ: 178.1 (s, C4), 163.8 (s, C2), 157.4 (s, C9), 144.0 (d, C7), 134.0 (s, C1′), 131.1 (s, C7′, C9′), 130.2 (d, C6′, C10′), 130.0 (d, C5), 128.8 (s, C2′, C14′), 127.3 (d, C3′, C13′), 126.1 (d, C4′, C12′), 125.7 (d, C5′, C11′), 125.6 (s, C10), 125.0 (d, C6), 124.3 (d, C8′), 118.5 (d, C8), 116.3 (d, C3). GC-MS m/z (rel. int.) = 322.27 [M]$^+$ (7), 321.25 (4), 305.25 (9), 202.15 (19), 200.16 (9), 189.11 (2), 120.06 (15), 92.08 (100).

2-(furan-2-yl)-4H-chromen-4-one (**3k**): brown yellow powder (yield 45%). $C_{13}H_8O_3$. Mp = 118–120 °C. IR (ATR diamond, cm$^{-1}$) = 3105 (C-H), 1651 (C=O), 1054 (C-O) cm$^{-1}$. $^1$H NMR (400 MHz, CDCl$_3$) δ: 8.21 (dd, $J_1$ = 7.9, $J_2$ = 1.5 Hz, 1H, H5), 7.68 (ddd, $J_1$ = 8.5, $J_2$ = 8.7, $J_3$ = 1.6 Hz, 1H, H7), 7.62 (dd, $J_1$ = 2.3, $J_2$ = 1.6 Hz, 1H, H4′), 7.49 (dd, $J_1$ = 8.4, $J_2$ = 1.0 Hz, 1H, H8), 7.41 (ddd, $J_1$ = 8.0, $J_2$ = 8.0, $J_3$ = 1.0 Hz, 1H, H6), 7.13 (d, 1H, H2′), 6.72 (s, 1H, H3), 6.60 (dd, $J_1$ = 3.5, $J_2$ = 1.7 Hz, 1H, H3′). $^{13}$C NMR (100 MHz, CDCl$_3$) δ: 177.8 (s, C4), 155.9 (s, C2), 155.2 (s, C9), 146.5 (s, C1′), 145.9 (d, C4′), 133.8 (d, C7), 125.8 (d, C5), 125.2 (s, C10), 124.3 (d, C6), 117.9 (d, C8), 113.1 (d, C3′), 112.6 (d, C3), 105.6 (d, C2′). GC-MS m/z (rel. int.) = 212.12 [M]$^+$ (11), 184.12 (7), 156.12 (2), 128.12 (9), 120.07 (24), 92.07 (100), 63.07 (94).

*2.4. In Vitro Antioxidant Activity assay*

2.4.1. DPPH Radical-Scavenging Assay

The radical-scavenging activity was performed, as described by Salazar-Aranda et al. [26], with slight modifications. Different concentrations of the sample were prepared with methanol by serial

dilutions. An aliquot of each dilution (0.5 mL) was mixed with 0.5 mL of a methanolic solution of 1,1-diphenyl-2-picrylhydrazyl (DPPH) (7.5 mg/250 mL). The mixtures were incubated at room temperature in the dark for 30 min. Using methanol as a blank, the spectrophotometric measurement was performed at 517 nm ($A_{517}$). The radical-scavenging activity was calculated as a percentage of DPPH decoloration using the following formula:

$$\text{DPPH (\%)} = [1 - (B/A)] \times 100 \tag{1}$$

Where A represents the A value of the control (DPPH solution) and B represents the sample. All of the determinations were performed in triplicate. For each compound, the percentage of DPPH decoloration was plotted against the concentration of each dilution. The concentration that is required to decrease the absorbance of DPPH by 50% was obtained by interpolation, from a linear regression analysis, and expresses the $EC_{50}$ value. Quercetin was used as a reference compound.

2.4.2. ABTS (2-2′-azino-bis-(3-ethylbenzothiazoline-6-sulfonate)) Radical-Scavenging Assay

ABTS radical cation (ABTS$^+$) scavenging assay was determined using the methodology developed by Re et al. and Kuskoski et al. [27,28] with slight modifications. ABTS was dissolved in water at 7 mM concentration. ABTS radical cation (ABTS$^+$) was produced by reacting ABTS stock solution with 2.45 mM potassium persulfate and then the mixture was kept in the dark at room temperature for 16-18 h before use. The ABTS$^+$ solution (150 µL) was diluted with methanol to obtain an absorbance at $0.7 \pm 0.02$, which is taken as initial absorbance. Of this solution, 980 µL was added to 20 µL of different concentrations of the sample. The mixture was stirred, incubated at room temperature for 30 min, and read at a wavelength of 754 nm, which was taken as the final absorbance. The results are shown as a percentage of inhibition using the following formula:

$$\text{\% of inhibition} = [(A1 - A2)/A1] \times 100 \tag{2}$$

Where A1 is the initial absorbance of the ABTS solution and A2 is the final absorbance of the ABTS solution in the presence of the sample. All of the determinations were performed in triplicate. For each compound, the percentage of inhibition was plotted against the concentration of each dilution. The concentration required to decrease the absorbance of ABTS by 50% was obtained by interpolation, from a linear regression analysis, and expresses the $EC_{50}$ value. Quercetin was used as a reference compound.

2.4.3. β-Carotene/linoleic Acid bleaching Assay

Heat-induced oxidation of an aqueous emulsion system of β-carotene and linoleic acid was used for testing the antioxidant activity that was described by Burda and Oleszek [29], with slight modifications. To 1 mL of β-carotene solution (0.2 mg/mL in chloroform) was added 20 µL linoleic acid and 200 µL of Tween 20. Once the chloroform was evaporated, 50 mL of distilled water saturated for 1 h with oxygen was added. An amount of 2.5 mL of the β-carotene emulsion was mixed with 0.35 mL of the corresponding compound solution in methanol (0.5 mg/mL) and the resulting mixture was shaken. The absorbance of the samples was measured on a spectrometer at 470 nm immediately after their preparation (t = 0 min), and at the end of the experiment (t = 120 min), during this time the samples were kept at 50 °C. The antioxidant activity (AA) was expressed as percentage of inhibition of β-carotene bleaching, as compared to the control, and calculated using the following formula:

$$\text{AA (\%)} = \{1 - [(AS0 - AS120)/(AC0 - AC120)]\} \times 100 \tag{3}$$

Where $As^0$ is the absorbance of the sample at 0 min, $As^{120}$ is the absorbance of the sample at 120 min, $Ac^0$ is the absorbance of the control sample at 0 min, and $Ac^{120}$ is the absorbance of the control sample at 120 min. α-tocopherol was used as a reference compound.

### 2.5. In Vitro Acetylcholinesterase Inhibitory Assay

Acetylcholinesterase activity was determined with the methodology that was reported by Adewusi (2011) [30]. In a 96-well plate was added 75 μL of Trizma-HCl buffer (50 mM, pH 8) and 75 μL of synthesized compound and serial dilutions were made (final well concentrations were between 150 and 2.4 μg/mL in the buffer, and DMSO at a final maximum concentration of 0.15%). Afterwards, to each well was added 25 μL of a buffer solution of ATCl 15 mM (final concentration of 1.5 mM) and 125 μL of a buffer solution of DTNB 3 mM (final concentration 1.5 mM). Absorbance was measured in a microplate reader (Multiskan FC, Thermo Scientific) at 405 nm every 45 s in three consecutive times. After that, to each well was added 25 μL of an acetylcholinesterase enzyme buffer solution at 2 U/mL enriched with bovine serum albumin 0.1 mg/mL (final well enzyme concentration 0.2 U/mL) and the absorbance was measured five consecutive times every 45 s. Six wells of each plate were employed without a compound to test, to serve as a 100% enzymatic activity control. The increase of absorbance due to spontaneous substrate hydrolysis was corrected subtracting the absorbance before the enzyme addition from the absorbance that was obtained after the enzyme addition. Inhibition percentage was calculated by the following equation

$$\text{inhibition } \% = 1 - (A\text{sample}/A\text{control}) \times 100 \tag{4}$$

Where $A_{\text{sample}}$ is the absorbance difference between time 0 and 225 s in presence of any compound to test or inhibitor, and $A_{\text{control}}$ is the absorbance difference between time 0 and 225 s of the 100% enzymatic activity control. Inhibitory concentration 50 ($IC_{50}$) was calculated by interpolation from the graphic of inhibition percentages in function of the employed concentrations. All of the experiments were performed by triplicate and galantamine was employed as a positive control, which is a drug that is currently used for Alzheimer disease treatment. [12]

### 2.6. Molecular Docking

The three-dimensional structures of the ligands were generated with UCSF Chimera 1.11.2 [31] through their SMILES string. The structures were energy-minimized using Chimera default parameters (which include MMTK, Amber, and Amber's Antechamber parameters) [32]. Rotatable bonds and atomic charges from the ligands were defined with AutoDock Tools 1.5.6. [33]. The crystallographic structure of the AChE receptor (PDB ID:1EVE) in complex with donepezil was downloaded from the Protein Data Bank (https://www.rcsb.org/). The preparation for the receptor with AutoDock Tools included remotion of the co-crystalized ligand and water molecules, hydrogen addition and charges calculation. For the docking, AutoDock Vina [34] was employed, with a grid box of 30 × 30 × 30 Å with center coordinates x = 2.27731, y = 63.7499 and z = 65.4998. Five docking runs with 10 poses each were made per ligand, with an exhaustiveness of 8. Visualizations of the docked poses for their analysis were made with Chimera. To validate the docking protocol, a re-docking of co-crystallized donepezil was made and the resulting pose was compared to the crystallographic one through their RMSD.

## 3. Results and Discussion

### 3.1. Synthesis of Chalcone and Flavone Analogs (2a–2k, 3a–3k)

Chalcones are usually synthesized by condensation reactions with acid or basic catalysis, even though recently a great number of new procedures for the synthesis of these molecules due to the great interest on their biological properties have appeared. Claisen–Schmidt reaction with basic conditions is the most widely synthesis reported in the literature, because of the easily process and efficiency in the product formation [35]. The substituted 2-hydroxychalcones **2a–2k** were prepared by the Claisen–Schmidt condensation of the respective substituted benzaldehyde **1a–1k** (1 eq) and 2-hydroxyacetophenone (1 eq) in the presence of NaOH in ethanol/water at room temperature by the known literature method (Scheme 1) [36]. This methodology allowed for the obtention of

eleven compounds with moderate to high yields, between 40–97%, being the highest one the yield for compound **2h**. The oxidative cyclization of 2-hydroxychalcones **2a–2k** to flavones **3a–3k** was performed using the classical iodine (1 mmol) in DMSO (Scheme 1) system. This method is particularly useful in flavone synthesis starting from 2-hydroxychalcones as it has been observed the iodine does not give secondary reactions despite the high temperature employed [37]. The corresponding flavones were successfully synthesized with yields between 40–85%.

**1a** $R_1 = R_2 = R_3 = R_4 = H$
**1b** $R_1 = OH, R_2 = R_3 = R_4 = H$
**1c** $R_1 = R_4 = H, R_2 = OMe, R_3 = OH$
**1d** $R_1 = OH, R_2 = R_3 = H, R_4 = NO_2$
**1e** $R_1 = R_2 = R_4 = H, R_3 = OMe$
**1f** $R_1 = R_3 = R_4 = H, R_2 = OMe$
**1g** $R_1 = R_4 = H, R_2 = R_3 = OMe$
**1h** $R_1 = R_2 = R_4 = H, R_3 = Cl$
**1i** $R_1 = R_2 = R_4 = H, R_3 = N(CH_3)_2$
**1j** 9-anthraldehyde
**1k** Furfuraldehyde

**Scheme 1.** Reaction Conditions: (**i**) NaOH 50%/ethanol-$H_2O$, room temperature, 12 h. (**ii**) $I_2$ – DMSO, reflux 130 °C, 30 min.

All of the compounds were characterized by IR, [1]H, and [13]C-NMR and mass spectroscopy; in the [1]H NMR spectra of the compounds **2a–k** can be noticed the hydroxyl proton in ring A between δ 12.63–13.18 ppm, due to the intramolecular hydrogen bond made with the carbonyl oxygen atom. The vinylic protons of the α,β-unsaturated system are present as doublets for $H_\alpha$ = 7.48–8.10 ppm and $H_\beta$ =7.69–8.89 ppm, being the coupling constants $JH_\alpha$-$JH_\beta$ = 14.9–15.8 Hz, which indicates a *trans* configuration for these protons. All of the aromatic protons were observed at their expected shifts, so as their coupling constants. In [13]C-NMR spectra, the carbonyl carbon is observed at δ 193.3–193.8 ppm, being the α- and β- carbons at 117.7–121.7 ppm and 129.7–145.8 ppm, respectively.

For the flavone type compounds **3a–k**, carbonyl α-proton H-3 is observed as a singlet with shifts of 6.69–7.12 ppm, the aromatic protons of the A and B rings were at their expected shifts and coupling constants. Regarding [13]C-NMR, the carbonyl carbon shifts to a lower frequency with δ 177.0–193.7 ppm, being the carbons C-2 and C-3 identified at 155.9–163.9 ppm and 104.5–116.3 ppm, respectively. The assignments were made by HSQC and HMBC experiments.

NMR spectra, including bidimensional ones, of selected chalcones and flavones can be observed in Figures S1–S24 in the supplementary materials.

## 3.2. Antioxidant Activity

To determine antioxidant activities of 2-hydroxychalcones **2a–2k** and flavone analogs **3a–3k**, heat-induced oxidation in a β-carotene and linoleic acid system, DPPH radical scavenging activity and ABTS radical cation decolorization assay were evaluated. The $EC_{50}$ values of compounds, quercetin and α-tocopherol obtained are shown in Table 1 and Figure 2.

**Table 1.** Antioxidant activity of synthesized compounds **2a–2k** and **3a–3k**.

| Compound | B-Ring Substituent | | | | Scavenging Activity (EC$_{50}$, µg/mL) | |
|---|---|---|---|---|---|---|
| | R1 | R2 | R3 | R4 | ABTS (2-2′-azino-bis-(3-ethylbenzothiazoline-6-sulfonate)) | DPPH |
| **2a** | | | | | $1.61 \times 10^4 \pm 5.73 \times 10^{-3}$ | $260 \pm 1.17 \times 10^{-2}$ |
| **2b** | OH | | | | $21 \pm 7.07 \times 10^{-5}$ | $8 \pm 1.24 \times 10^{-4}$ |
| **2c** | | OMe | OH | | $164 \pm 4.95 \times 10^{-4}$ | $19 \pm 1.18 \times 10^{-3}$ |
| **2d** | OH | | | NO$_2$ | $564 \pm 4.24 \times 10^{-4}$ | $23 \pm 9.76 \times 10^{-4}$ |
| **2e** | | | OMe | | $341 \pm 2.83 \times 10^{-4}$ | $100 \pm 2.57 \times 10^{-3}$ |
| **2f** | | OMe | | | $487 \pm 6.93 \times 10^{-3}$ | $170 \pm 4.10 \times 10^{-3}$ |
| **2g** | | OMe | OMe | | $1.14 \times 10^{-2} \pm 7.42 \times 10^{-7}$ | $5 \times 10^4 \pm 7.76 \times 10^{-1}$ |
| **2h** | | | Cl | | $5.23 \times 10^3 \pm 9.90 \times 10^{-4}$ | $15 \pm 7.43 \times 10^{-4}$ |
| **2i** | | | | N(CH$_3$)$_2$ | $6.96 \times 10^{-2} \pm 2.12 \times 10^{-7}$ | $9.3 \pm 4.95 \times 10^{-5}$ |
| **2j** | | Anthracene | | | $548 \pm 3.54 \times 10^{-3}$ | $1.7 \pm 6.82 \times 10^{-5}$ |
| **2k** | | Furan | | | $79 \pm 6.36 \times 10^{-4}$ | $5.4 \times 10^{-3} \pm 6.50 \times 10^{-8}$ |
| **3a** | | | | | $2.53 \times 10^4 \pm 4.24 \times 10^{-3}$ | $170 \pm 1.94 \times 10^{-5}$ |
| **3b** | OH | | | | $0.3 \pm 3.54 \times 10^{-5}$ | $0.1 \pm 1.39 \times 10^{-6}$ |
| **3c** | | OMe | OH | | $497 \pm 1.98 \times 10^{-4}$ | $9 \pm 4.60 \times 10^{-5}$ |
| **3d** | OH | | | NO$_2$ | $155 \pm 1.11 \times 10^{-3}$ | $6.2 \pm 1.52 \times 10^{-4}$ |
| **3e** | | | OMe | | $3.00 \times 10^3 \pm 1.21 \times 10^{-3}$ | $30 \pm 3.61 \times 10^{-3}$ |
| **3f** | | OMe | | | $112 \pm 2.90 \times 10^{-4}$ | $39 \pm 1.58 \times 10^{-3}$ |
| **3g** | | OMe | OMe | | $861 \pm 2.55 \times 10^{-4}$ | $410 \pm 6.85 \times 10^{-3}$ |
| **3h** | | | Cl | | $2.6 \times 10^3 \pm 1.06 \times 10^{-3}$ | $7.4 \pm 1.09 \times 10^{-5}$ |
| **3i** | | | | N(CH$_3$)$_2$ | $1.9 \times 10^{-3} \pm 6.51 \times 10^{-8}$ | $0.25 \pm 2.14 \times 10^{-5}$ |
| **3j** | | Anthracene | | | $1.07 \times 10^5 \pm 3.99 \times 10^{-4}$ | $1 \times 10^{-8} \pm 1.61 \times 10^{-12}$ |
| **3k** | | Furan | | | $1.26 \times 10^3 \pm 7.78 \times 10^{-4}$ | $0.1 \pm 7.85 \times 10^{-6}$ |
| *Quercetin | | | | | $50 \pm 4.1 \times 10^{-4}$ | $3 \pm 2 \times 10^{-4}$ |

\* Served as reference compound. Values are mean $\pm$ SD, n = 2.

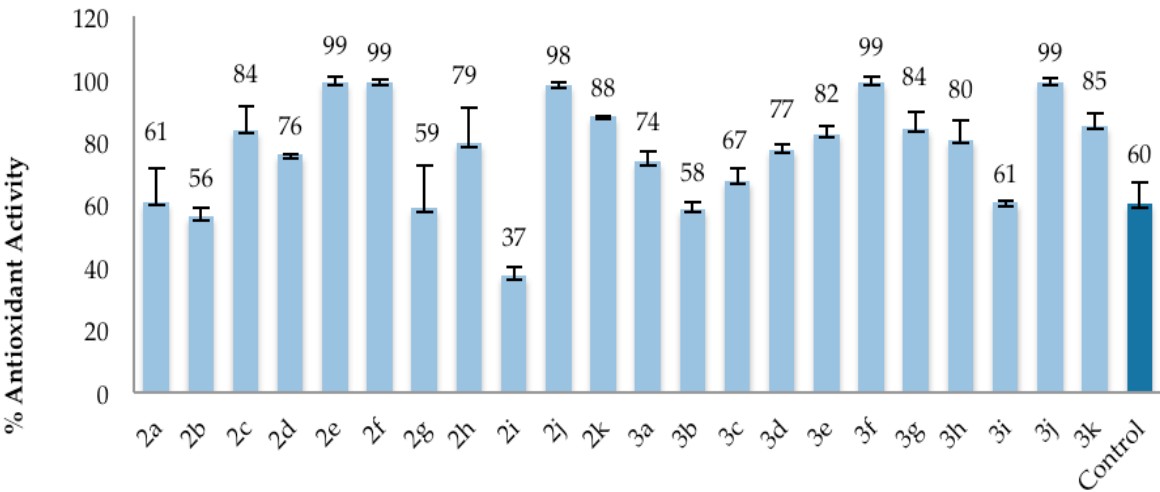

**Figure 2.** Percentage of antioxidant activity in β-carotene/linoleic acid bleaching assay of chalcone (**2a–2k**) and flavone analogs (**3a–3k**). α-tocopherol served as reference compound. Bars are mean $\pm$ SD, n = 2.

The results showed that most of the compounds exhibited high activity with DPPH and β-carotene assay and weak activity in ABTS; additionally, the flavone-type derivatives showed a better antiradical and antioxidant activity as compared to their chalcone precursors. Structure-antioxidant activity relationships of chalcones and flavones suggest this activity is related to a series of different mechanisms, like free radical neutralization, hydrogen donation, singlet oxygen quenching, and metal chelation [38].

The high double bond conjugation of chalcone and flavonoid systems, just as the presence of a C2-C3 double bond and a carbonyl in C-4 position, have been reported as structural characteristics for the antioxidant activity [39], being these incorporated in the basic skeleton of all the synthesized compounds.

Bearing this in mind, molecules **2a** and **3a** have no substituents in their structures, having the lowest activity of all the generated analogs, therefore it is possible to infer that the addition of at least one substitution favors the antioxidant potential.

In chalcone type molecules, the antioxidant properties are vastly influenced by the two aryl groups and their substitution patterns. The presence of hydroxyl groups to improve the antioxidant activity due to its easy conversion to phenoxy radicals by the hydrogen transfer mechanism is considered to be indispensable [40].

Hydroxyl group presence was evaluated in the **2b** (*o*-OH), **2c** (*p*-OH, *m*-OMe), and **2d** (*o*-OH, *m*-NO$_2$) analogs; despite that the three had great activity in the tested techniques (see Table 1), **2b** showed better results when compared to the other analogs in the antiradical techniques DPPH and ABTS, with an EC$_{50}$ of 8 μg/mL and 21 μg/mL respectively, surpassing quercetin as the control used in ABTS (EC$_{50}$ of 50 μg/mL). This indicates that in the radical reduction mechanism by hydrogen donation (like DPPH and ABTS), the addition of other substitutions different from hydroxyl groups diminishes activity, especially using electron-attractor groups as NO$_2$ [41]. A similar behavior is appreciated in the results of flavones **3b**, **3c**, and **3d**, with **3b** showing good activity with an EC$_{50}$ of 0.1 μg/mL in DPPH and 0.3 μg/mL in ABTS, in both situations surpassing the positive control; again, it was observed that the presence of electron-attractor groups lowers the activity. Analogs **2h** and **3h** with *p*-Cl substitutions did not show considerable activity when compared to the controls.

The influence of electron-donor groups, like methoxyl, can be analyzed with the **2e** (*p*-OMe), **2f** (*m*-OMe), and **2g** (*p*,*m*-OMe) analogs; the presence of these groups did not favor the activity, getting concentrations of 100, 170, and 5 × 10$^4$ μg/mL, respectively, in DPPH assay, having a similar behavior of the flavones **3e**–**g**, with EC$_{50}$ of 30, 39, and 410 μg/mL. These results may appear to contrast with the reported by many authors where methoxyl groups are considered an important factor for the antioxidant activity [42,43], however, this is the case where methoxyl groups help with the lipophilicity of the molecule, for example, when the antioxidant activity is measured by the lipid peroxidation activity assay [39]. When employing DPPH test, the transfer of acid protons is required; this implies that just as in the case of the chalcones, we consider indispensable the presence of at least one hydroxyl group, explaining the lack of scavenging activity for the synthesized compounds with only methoxyl groups (**3e**–**g**).

Molecules **2i**–**3i**, both with *p*-dimethylamino substituent showed excellent results with an EC$_{50}$ of 9.3 and 0.25 μg/mL, respectively, in the DPPH assay, having an EC$_{50}$ of 6.9 × 10$^{-2}$ and 1.9 × 10$^{-3}$ μg/mL in ABTS; substituents that increase the electronic density, like dimethylamine groups, show potent antioxidant activity that is attributed to the resonance effect involving the pair of electrons of the heteroatom [44].

The β-carotene decolorization assay brings an outlook to the capacity of an antioxidant to inhibit the lipid peroxidation initiated by the hydrogen abstraction or addition of oxygen radical [45]. The chalcones **2c** and **2d** with hydroxyl substitution showed good activity in this technique with percentages of activity between 76–84%, while the flavones with the same substitution showed results between 67–77%. Undoubtedly, the presence of methoxyl groups in the analogs B ring favored the antioxidant activity, the chalcones **2e** and **2f** exhibit 99% of antioxidant activity, while the flavones **3e**, **3f**, and **3g** present results of 82, 99 and 84% respectively, all of them better than α-tocopherol (60%). This allows for determining that the presence of electron-donor groups in the B ring benefits this activity [46,47].

From all the synthesized analogs, it is important to highlight the results obtained for **2j**, **2k**, **3j**, and **3k**, which have anthracene and furan substitutions, while they presented antiradical activity at very low concentrations in the DPPH assay, with EC$_{50}$ of 1.7 and 5.4 × 10$^{-3}$ μg/mL for **2j** and **2k**, and 1 × 10$^{-8}$ and 0.1 μg/mL for **3j** and **3k**, being far better than the control quercetin (3 μg/mL).

These same four molecules showed a high antioxidant activity in the β-carotene technique, with results of 98% and 88% for the **2j** and **2k** chalcones, while the **3j** and **3k** flavones had 99% and 85%, respectively, surpassing the 60% antioxidant activity of α-tocopherol. It is known that the oxidation

of polycyclic aromatic hydrocarbons, such as anthracene generates anthraquinone type products, with 9,10-anthraquinone and 9-hydroxyanthrone being the more abundant [48], the basic skeleton for anthraquinones and anthrone, which provides them the capacity to act as electron acceptors (electrophiles); furthermore, their high conjugation gives them great stability (by resonance means), potentiating the antioxidant activity [49,50]. On other hand, the oxidation of furans gives as a result oxyfuranones in the case of monosubstitutions, or diketones in the case of 2,5-disubstituted furans [51]; Y. Sugiyama et al. reported the importance of diketonic systems in the antioxidant activity, both in their keto or enol form [52].

### 3.3. Acetylcholinesterase Inhibitory Assay

The acetylcholinesterase inhibitory activity of the synthesized compounds was evaluated employing Adewusi adaptation of Ellmans spectrophotometrical assay [30], using galantamine as a reference. The results for these experiments are shown with their $IC_{50}$ values on Table 2. The compounds that were chosen for screening can be divided into two groups based on their structural features: chalcones and flavones.

**Table 2.** Acetylcholinesterase inhibitory activity of synthesized compounds **2a**–**2k** and **3a**–**3k**.

| Compound | $IC_{50}$ (μg/mL) | Compound | $IC_{50}$ (μg/mL) |
|:---:|:---:|:---:|:---:|
| **2a** | > 150 | **3a** | > 150 |
| **2b** | > 150 | **3b** | 61.2 ± 1.39 |
| **2c** | 52.7 ± 11.98 | **3c** | 78.4 ± 1.92 |
| **2d** | 21.5 ± 2.61 | **3d** | 26.8 ± 5.91 |
| **2e** | > 150 | **3e** | 43.5 ± 4.31 |
| **2f** | > 150 | **3f** | 60 ± 9.96 |
| **2g** | > 150 | **3g** | 43.5 ± 2.52 |
| **2h** | > 150 | **3h** | > 150 |
| **2i** | 80.4 ± 5.97 | **3i** | > 150 |
| **2j** | > 150 | **3j** | 77.6 ± 31.89 |
| **2k** | 66.4 ± 6.15 | **3k** | > 150 |
| *Galantamine | | 0.574 ± 0.07 | |

* Served as reference compound. Values are mean ± SD, n = 3.

Chalcones are 1,3-diphenyl-2-propene-1-one systems, in which two aromatic rings are linked by a three carbon α, β-unsaturated carbonyl system. The compounds with this backbone have been reported to possess various biological activities [53]. The $IC_{50}$ values of these compounds are listed in Table 2. Hasan et al. [54] reported that hydroxyl groups in ortho position of the A ring of chalcones, are an important structural element in the AChE inhibitory activity; however, in our study all the synthesized chalcones had the *o*-OH substitution in their A ring, and most of them displayed no inhibition effect in the tested concentration scale. Compounds **2c**, **2i**, and **2k** showed an inhibitory effect with $IC_{50}$ values of 52.7, 80.4, and 66.4 μg/mL, respectively, in contrast to the rest of molecules that presented results above 150 μg/mL.

On the other hand, results of the flavone group **3a**–**3k** that are based upon a fifteen-carbon skeleton consisting of two benzene rings (A and B) linked via a heterocyclic pyrane ring (C) (Figure 1) indicate that they were more active than the chalcone group. Only the compounds **3a**, **3h**, **3i**, and **3k** presented poor inhibition as compared to the rest of flavones that, in the majority, have hydroxyl or methoxyl groups in their B ring, where compounds **3e**, **3g**, **3f**, and **3b** outstand with an $IC_{50}$ of 43.5, 43.5, 60.0, and 61.2 μg/mL, respectively; however, compound **3d** with *o*-OH, *m*-$NO_2$ in the B ring was found to be the best flavone inhibitor of AChE, having an $IC_{50}$ of 26.8 μg/mL.

It has been acknowledged that hydroxyl and methoxyl groups in the A and B rings of flavones are deeply related to bioactivity and their capacity to protein bonding, being the hydroxyl groups that are favorable for the flavonoids interaction with acetylcholinesterase by hydrogen bonds formation in the enzyme peripheral anionic site (PAS) [55], which would explain the activity showed by compounds **3b**,

**3c**, and **3d** (61.2, 78.4, and 26.8 μg/mL). On other hand, the presence of methoxyl groups in flavonoids favors the inhibition effect due to the interaction with residue Trp279, located in the entrance to the active site (in the PAS), this could explain the results obtained for analogs **3e–3g**.

While we can notice a general improvement of the results with the presence of hydroxyl and methoxyl groups in the chalcones B ring, Sukumaran et al. [56] mentions that, in 2′-hydroxychalcones, the AChE inhibition is generally favored with halogens in the B ring. Nevertheless, their chalcone with chlorine in *p*-position of the B ring did not showed significant activity, molecule that corresponds with our compound **2h**, from which we neither detect activity in the tested concentration scale.

As can be noticed, chalcone **2d** and flavone **3d**, both with *o*-OH and *m*-NO$_2$ substitutions, were the most active compounds with IC$_{50}$ values of 21.5 and 26.8 μg/mL, respectively, these results suggest that the presence of the nitro group in both molecules enhances the inhibitory activity; it has been previously reported that the introduction of basic or permanently charged nitrogen atoms and aromatic systems, are common structural characteristics in molecules for their interaction with the binding sites of acetylcholinesterase [57].

### 3.4. Molecular Docking Analysis

For the molecular docking studies, the compounds with the best in vitro results **2d** and **3d** were selected in order to gain insight of their interactions with the AChE receptor and explain their biological results. AutoDock Vina was selected to perform the in silico analysis, using the crystallographic structure of the Pacific electric ray *Torpedo californica* AChE (TcAChE) available from Protein Data Bank (PDB ID:1EVE) as its similarity with human AChE allows for employing it in docking studies [58]. Having selected 1EVE, a docking grid was generated in the active site region defined by the co-crystallized receptor's ligand donepezil. To assure effectiveness of the analysis, it was necessary to validate the method; for this, a re-docking of the co-crystallized donepezil was realized with the same parameters intended to use for our ligands. We compared the best result pose for donepezil with the co-crystallized one obtaining a RMSD of 1.1, which, being lower than 1.5–2 Å, is considered to be a successful analysis, suitable for predicting ligand poses [59].

In Table 3, the resulting scores of the docking for compounds **2d** and **3d** with the inhibition constants can be observed, which AutoDock calculates by the formula:

$$Ki = \exp(\Delta G \times 1000 \text{ Rcal} \times TK) \tag{5}$$

$\Delta G$ is the docking binding energy, R$_{cal}$ value is 1.98719, and T$_K$ value is 298.15. In the in vitro results, chalcone **2d** was the most active synthesized compound with an IC$_{50}$ of 21.5 μg/mL, followed by the flavone **3d** (26.8 μg/mL); accordingly, the docking gave a better score for **2d** ($-10.2$ kcal/mol) against **3d** ($-9.78$ kcal/mol). The more negative the docking score (binding energy value), the better affinity the ligand has for the receptor.

**Table 3.** Docking scores for compounds **2d** and **3d**.

| Compound | Binding Energy (kcal/mol) | Inhibition Constant (nM) | Principal Residues Interactions |
|:---:|:---:|:---:|:---:|
| **2d** | $-10.2$ | 33.37 | Trp84, Tyr334, Tyr130 |
| **3d** | $-9.78$ | 67.79 | Trp84, Tyr334, Tyr130 |

Analysis of the poses generated showed us that our compounds interact with AChE mostly by π-π interactions with the active site residues.

The active site of AChE consists in a cavity or gorge in the receptor surface that is approximately 20 Å deep; besides the catalytic triad of the enzyme (Ser200, His440, and Glu327), which is located at the bottom of the gorge, several residues have been identified as important zones for the interaction of ligands, such as Trp84, Phe330, Glu199, and Gly441 that make the central anionic site (CAS), and

Trp279, Tyr334, Tyr121, and Tyr70 at the peripheral anionic site (PAS) [60,61]. Both **2d** and **3d** poses are observed nearer to the CAS region, as seen in the Figure 3.

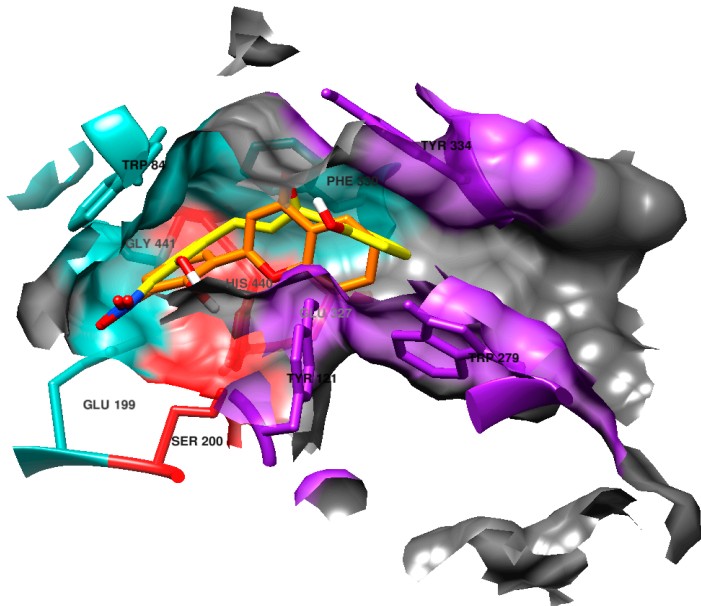

**Figure 3.** Gorge of 1EVE depicting compounds **2d** (yellow) and **3d** (orange) position and the catalytic triad (red), central anionic site (CAS) (blue), and peripheral anionic site (PAS) (purple).

The best pose generated for **2d** does not form hydrogen bonds to the nearest residues; although, one oxygen of the nitro group is near the hydrogen of Tyr130, at only 2.7 Å, the angle between them does not allow a proper hydrogen bond. A small turn of the nitro or hydroxyl groups would be needed for this to happen, but a weak interaction between them might be helping the interaction of **2d** with this residue. The clearer interactions for **2d** appear with Trp84, a residue that has been reported as an important one for ligand interactions (especially with quaternary groups from acetylcholine and other compounds) [58,62]. In this case, Trp84 is near the positively charged nitrogen of the nitro group, although its position appears to be almost over the B ring of the chalcone, allowing for π-π stacking at 3.7 Å. Another π-π stacking can be observed between the A ring of **2d** and Tyr334 in the PAS region, with 4 Å among the aromatic rings (Figure 4). The side of chalcone **2d** with the A ring rests in the mostly hydrophobic zone created by Trp334, Phe331, and Phe330, while the side of the B ring points to a less hydrophobic area.

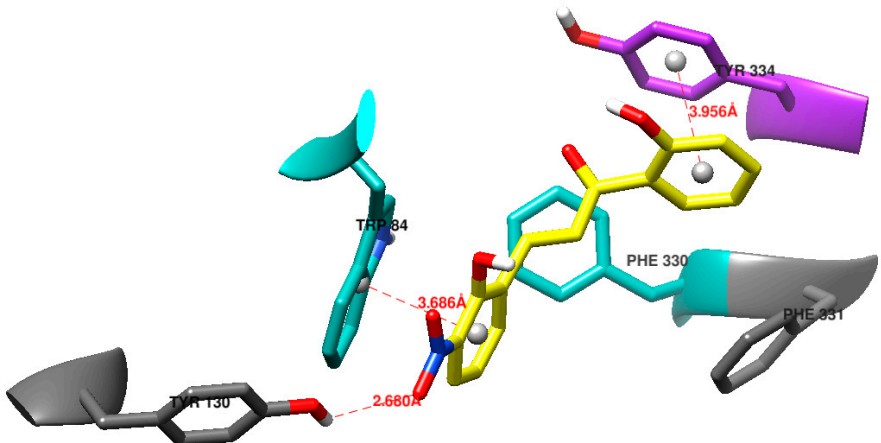

**Figure 4.** Chalcone **2d** and its interactions with nearby residues.

The case for the best generated pose of flavone **3d** is very similar to its chalcone counterpart. Although a formal hydrogen bond is not detected by the Chimera software due to the angle between them, one oxygen of the nitro group is near the hydrogen of Tyr130, at 2.6 Å, so a weak interaction among these atoms could be possible (Figure 5). The nitro group is near to Trp84, but the B ring appears to have a better interaction with this residue in form of π-π stacking at a distance of 4 Å. The A ring has proximity to Tyr334, even though with the new ring formation the flavone **3d** has a more rigid position for the A ring, which now appears to have with its edge a perpendicular π-π interaction with Tyr334 at 3.6 Å. The C ring is oriented to Phe330 with a distance among them of 4.4 Å. As the chalcone **2d** did, the B ring with the nitro group of the flavone **3d** is oriented to the bottom of the gorge, in a less hydrophobic area, while the rest of the structure is towards the hydrophobic zone of Trp334, Phe331, and Phe330, as can be seen in Figure 6.

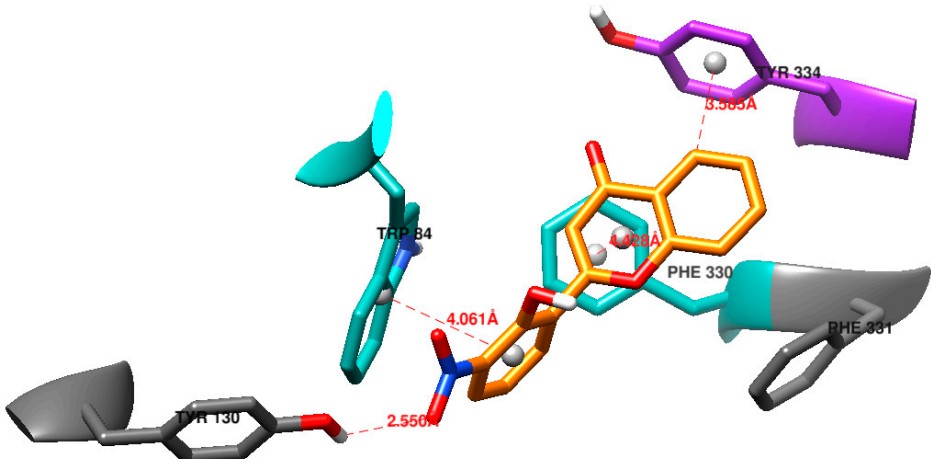

**Figure 5.** Flavone **3d** and its interactions with nearby residues.

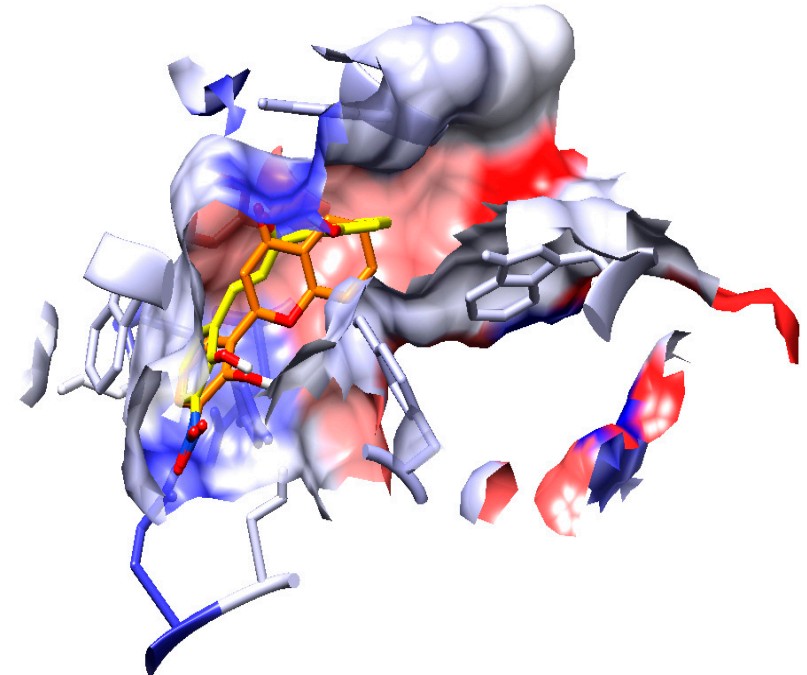

**Figure 6.** Compounds **2d** (yellow) and **3d** (orange) at the gorge of TcAChE. The color of the surface of the receptor indicates hydrophilicity (blue) passing by white to red (hydrophobicity).

## 4. Conclusions

In conclusion, for the three analyzed techniques, the flavone type analogs showed a better antioxidant profile when compared to the chalcones, however this does not mean that the latter were inactive since many of them resulted highly active. Surprisingly, the compounds **2j, 2k**, **3j**, and **3k** presented a high potential as antioxidant agents, this could be due to the quinone and oxyfuranone type products that are generated after their oxidation. As previous reports have stated, the presence of hydroxyl groups favors the antiradical activity mediated by hydrogen donation, as it could be noticed that compounds **2b** and **3b** surpassed the positive controls in both DPPH and ABTS assays; in the β-carotene technique, the electron-donor substitutions were the most active.

On the other hand, in the AChE inhibition assay, the better results from the chalcone and flavone families (compounds **2d** and **3d**) share the nitro functional group, with IC$_{50}$ values of 21.5 and 26.8 μg/mL, respectively. The docking results suggest that the principal interactions compounds **2d** and **3d** have with AChE active site are mostly π-π stackings. This can be observed with residues Trp84 and Tyr334 from the CAS and PAS, which are known binding zones of the enzyme.

Although further in vivo testing must be performed, our results represent an important step towards the identification of improved antioxidants and acetylcholinesterase inhibitors.

**Supplementary Materials:** The following are available online at http://www.mdpi.com/2076-3417/9/3/410/s1, Figure S1: $^{1}$H-NMR Compound **2e**, Figure S2: $^{13}$C-NMR Compound **2e**, Figure S3: COSY-Compound **2e**, Figure S4: HSQC-Compound **2e**, Figure S5: HMBC-Compound **2e**, Figure S6: $^{1}$H-NMR Compound **3e**, Figure S7: $^{13}$C-NMR Compound **3e**, Figure S8: COSY Compound **3e**, Figure S9: HSQC Compound **3e**, Figure S10: HMBC Compound **3e**, Figure S11: $^{1}$H-NMR Compound **2k**, Figure S12: $^{13}$C-NMR Compound **2k**, Figure S13: $^{1}$H-NMR Compound **3k**, Figure S14: $^{13}$C-NMR Compound **3k**, Figure S15: $^{1}$H-NMR Compound **2h**, Figure S16: $^{13}$C-NMR Compound **2h**, Figure S17: $^{1}$H-NMR Compound **3h**, Figure S18: $^{13}$C-NMR Compound **3h**, Figure S19: $^{1}$H-NMR Compound **2d**, Figure S20: $^{13}$C-NMR Compound **2d**, Figure S21: $^{1}$H-NMR Compound **2j**, Figure S22: $^{13}$C-NMR Compound **2j**, Figure S23: $^{1}$H-NMR Compound **3j**, Figure S24: $^{13}$C-NMR Compound **3j**.

**Author Contributions:** Conceptualization, I.C.-G. supervised the whole study; L.D.-R. and A.E.-C. carried out the chemical synthesis; L.D.-R. and R.H.-M. performed the chemical assay, I.A.R.-E. and D.C.-V. performed the spectroscopic data; M.A.R. collaborated in the discussion and interpretation of the results; L.D.-R., A.E.-C. and V.G.-G. wrote the manuscript; R.S.-A. and N.W.d.T. performed the enzymatic assay; A.E.-C and M.A.R. carried out the in silico study. All authors read and approved the final manuscript.

**Acknowledgments:** We gratefully acknowledge the Facultad de Ciencias Químicas e Ingeniería, Universidad Autónoma de Baja California for the financing given for the realization of this project. Additionally, to the Consejo Nacional de Ciencia y Tecnología (CONACYT) for ITT NMR facilities (Grant INFR-2011-3-173395).

**Conflicts of Interest:** The authors declare no conflict of interest.

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
