# Peer review of "Synthesis, Biological Evaluation and Docking Studies of Chalcone and Flavone Analogs as Antioxidants and Acetylcholinesterase Inhibitors"

_applsci, doi:10.3390/app9030410_

Round 1

Reviewer 1 Report

The authors of the manuscript "Synthesis, biological evaluation and docking studies of chalcone and flavone analogs as antioxidants and acetylcholinesterase inhibitors" described the synthesis and biological evaluation of series of chalcone and flavone derivatives. The manuscript was really well written. Length of the paper is acceptable and data were excellent presented in tables and figures. Although some minor corrections must be included.

1. The authors must enhanced the discussion in the synthetic part as well as in the biological part. The previous synthesis of the chalcone and flavone derivatives must be mentioned and compare with their method (pros and cons).

2. In the same sense, there are several times that some of the compounds i.e. 2a, 3a, 3b.. etc,  had been evaluated as antioxidants/acetylcholinesterase inhibitors see (Med. Chem. Res 2018, 27(2), 520-530; Pharm. Biol. 2012, 50(2), 239-246; etc) the authors must do a deep search comparing and discussing the different values reported.

3. If there is a newly synthesized derivative, the authors must highlight.

4. I missed the NMR data for the compound 3d?

Currently, I do recommend this manuscript for publication in Applied Sciences after improvement of the discussion part.

Author Response

Thank you for the review you gave to our manuscript, and your observations of it. Please find attached a document containing our responses.

Thanks again for your support. 

            The authors

Reviewer 2 Report

Chalcone and flavanone derivatives are common known from their broad spectrum of biological activity, what is well documented in literature. Particularly interesting are results of their activity as potential inhibitors of acetylcholinesterase. This publication seems to be within the scope of journal. However it needs several corrections to be more acceptable for publication.

1. In introduction and in conclusion, short explanation of research novelty is needed.

2. In figure 1 carbon atoms should be numbered and also in structure of first compound in supplementary information.

3. In Materials and Methods detailed description of GC-MS method is necessary (carrier gas, detailed information about capillary column, kind of detector, temperature of detector and temperature of injector, temperature programme, etc.).

4. In Materials and Methods detailed information about used eluent in column chromatography should be given.

5. The numbering of carbon atoms in Figure 1 should be used to assign NMR signals to concrete protons.

6. Line 367 “trans” should be in italic.

7. Line 416 What is the reason, that in this research methoxyl groups are not an important factor for the antioxidant activity of flavone derivatives?

8. Line 448, line 452, line 459: Year of publication should be removed.

9. In text of manuscript should be emphasized that galantamine is a drug used in treatment of Alzheimer disease.

10. Obtained results of AChE inhibition should be discussed with results described in publication of Sukumaran et al. Molecules, 2016, 21, 955.

11. References should be formatted according to the requirements of the journal, for example, DOI number instead of CrossRef.

Author Response

Thank you for the review you gave to our manuscript, and your observations of it. Please find attached a document containing our responses.

Thanks again for your support. 

The authors.

Round 2

Reviewer 2 Report

The authors corrected publication carefully. I think that manuscript in present form is suitable for publication.